# 7-Hydroxycoumarin Induces Vasorelaxation in Animals with Essential Hypertension: Focus on Potassium Channels and Intracellular Ca^2+^ Mobilization

**DOI:** 10.3390/molecules27217324

**Published:** 2022-10-28

**Authors:** Rafael L. C. Jesus, Isnar L. P. Silva, Fênix A. Araújo, Raiana A. Moraes, Liliane B. Silva, Daniele S. Brito, Gabriela B. C. Lima, Quiara L. Alves, Darizy F. Silva

**Affiliations:** 1Laboratory of Cardiovascular Physiology and Pharmacology, Federal University of Bahia, Salvador 40110-902, Brazil; 2Gonçalo Moniz Institute, Oswaldo Cruz Foundation—FIOCRUZ, Salvador 40296-710, Brazil

**Keywords:** 7-hydroxycoumarin, umbelliferone, hypertensive rats, isometric records, vasorelaxation, superior mesenteric artery superior

## Abstract

Cardiovascular diseases (CVD) are the deadliest noncommunicable disease worldwide. Hypertension is the most prevalent risk factor for the development of CVD. Although there is a wide range of antihypertensive drugs, there still remains a lack of blood pressure control options for hypertensive patients. Additionally, natural products remain crucial to the design of new drugs. The natural product 7-hydroxycoumarin (7-HC) exhibits pharmacological properties linked to antihypertensive mechanisms of action. This study aimed to evaluate the vascular effects of 7-HC in an experimental model of essential hypertension. The isometric tension measurements assessed the relaxant effect induced by 7-HC (0.001 μM–300 μM) in superior mesenteric arteries isolated from hypertensive rats (SHR, 200–300 g). Our results suggest that the relaxant effect induced by 7-HC rely on K^+^-channels (K_ATP_, BK_Ca_, and, to a lesser extent, K_v_) activation and also on Ca^2+^ influx from sarcolemma and sarcoplasmic reticulum mobilization (inositol 1,4,5-triphosphate (IP3) and ryanodine receptors). Moreover, 7-HC diminishes the mesenteric artery’s responsiveness to α1-adrenergic agonist challenge and improves the actions of the muscarinic agonist and NO donor. The present work demonstrated that the relaxant mechanism of 7-HC in SHR involves endothelium-independent vasorelaxant factors. Additionally, 7-HC reduced vasoconstriction of the sympathetic agonist while improving vascular endothelium-dependent and independent relaxation.

## 1. Introduction

Cardiovascular diseases (CVD) are and will continue to be among the most significant health problems worldwide [1]. Hypertension remains the most prevalent common chronic disease risk factor for CVD. According to the Global Hypertension Practice Guideline 2020 from the International Society of Hypertension, hypertension accounts for 10.4 million deaths annually [2]. Additionally, hypertension comprises about 9% of global disability-adjusted life years [1,3]. However, due to the wide range of treatment options for hypertension, many patients do not achieve optimal blood pressure (BP) control. Thus, hypertensive patients who have not achieved treatment-mediated BP control using regularly available antihypertensive drugs at inadequate doses may present a condition known as resistant hypertension.

Resistant hypertension was defined by the European Guidelines (2018) [4] as a persistent increase in BP that remains above healthy levels, despite using three or more different classes of antihypertensive medications, one of which is a diuretic administered at the best-tolerated doses. As much as possible, the initial three-drug regimen should be standardized, including a blocker of the renin–angiotensin system, specifically an angiotensin-converting enzyme inhibitor or an angiotensin receptor blocker; a long-acting calcium channel blocker, most commonly amlodipine; and a long-acting thiazide-like diuretic, preferably chlorthalidone or indapamide [5]. In addition, the American Heart Association (2018) [6] extended this definition to patients with four or more antihypertensive medications, irrespective of their BP level. However, despite the fact that these definitions are currently well accepted, several observations have challenged these definitions, since patients represent a heterogeneous population with highly variable morbid risks, which depending on genetic, environmental, cultural, and social characteristics [7]. As the number of prescribed drugs increases and, consequently, the dosage schedule becomes more complex, adherence to treatment decreases. Therefore, the search for drugs with different therapeutic targets for the treatment of hypertension seems to be interesting [5]. Thus, searching for new therapeutic strategies to reduce the morbidity and mortality induced by hypertension and CVD is necessary.

Coumarins are a natural product and exhibit various biological effects, such as anti-HIV, antimicrobial, anti-inflammatory, anticancer, anticonvulsant, and MAO-inhibiting properties [8]. Additionally, several coumarins have known vasorelaxant activities [9,10,11]. 7-Hydroxycoumarin (7-HC) is a seven-hydroxylated coumarin with pharmacological properties, such as antifungal, antinociceptive, anti-inflammatory, and hepatoprotective effects [12,13,14,15]. However, few studies have demonstrated the action of 7-HC in the cardiovascular system. Baccard and colleagues (2000) [16] showed that 7-HC was described as promoting increased coronary blood flow and causing positive inotropic effects in the rat heart and had a direct vasorelaxant effect on the coronary arteries. Furthermore, Jagadeesh and colleagues (2016) [17] showed that 7-HC protects against isoproterenol-induced myocardial infarction in rats through free radical scavenging.

Previous studies in our laboratory, developed by Alves and colleagues (2020) [18], demonstrated, for the first time, that 7-HC was found to be nontoxic in H9 c2 cardiomyocytes cells and exerts a vasorelaxant effect in the superior mesenteric artery of Wistar rats. This effect would involve the inhibition of intracellular Ca^2+^ mobilization and the participation of potassium channels, which appear to be activated by 7-HC in mesenteric artery rings, particularly K_v_, BK_Ca_, K_ATP_, and K_ir_, leading to vasodilation.

Despite our work elucidating 7-HC’s mechanisms of action in the cardiovascular system, the main limitation of the study was that we could not infer the effects of 7-HC in a model of essential hypertension, since the hypertensive condition alters numerous signaling pathways. Therefore, we aimed to investigate the vascular effect of 7-HC and its cardiovascular activities in spontaneously hypertensive rats (SHR), specifically to describe the mechanisms of action involved in the observed responses.

## 2. Results

### 2.1. 7-HC Induces Endothelium-Independent-Relaxation in the Mesenteric Artery in Hypertensive Rats

7-HC (0.001 μM to 300 μM) induced similar concentration-dependent relaxation in pre-contracted (Phe, 1 μM) mesenteric artery rings from SHR, with or without the endothelium (effect (300 μM) = 83.2 ± 4.4%, (*n* = 6); effect (300 μM) = 100.4 ± 9.8%, (*n* = 6)) (Figure 1A,B). Furthermore, we also demonstrated that the contraction induced by Tyrode’s solution containing 80 mM KCl before (0.78 ± 0.07% (*n* = 5)) and after (1.01 ± 0.08% (*n* = 5)) administration of 7-HC did not show a statistically significant difference, suggesting that 7-HC can reversibly interact with cell receptors and also does not induce tissue damage. Interestingly, the vasorelaxation induced by 7-HC in mesenteric artery rings without endothelium from SHR (effect (10 μM) = 20.7± 4.7 (*n* = 6)) was significantly attenuated compared with Wistar rats (effect (10 μM) = 43.1 ± 5.7 (*n* = 7)) at the 10 µM concentration only (Appendix A). This suggests that complications arising from hypertension may influence 7-HC-induced vasorelaxation.

### 2.2. The Vasorelaxation Induced by 7-HC Involves K^+^ Channel Activation in Hypertensive Rats

In endothelium-denuded rings, 7-HC (0.001 μM to 300 μM) inhibited the sustained tonic contraction induced by Tyrode’s solution containing 80 mM KCl (effect (300 μM) = 63.8 ± 3.9%, (*n* = 6)) in a concentration-dependent manner. The response induced by 7-HC was significantly reduced (*p* < 0.01) compared with rings contracted with Phe (1 μM) (effect (300 μM) = 100.4 ± 9.8%, (*n* = 6)) (Figure 2A). However, arteries pre-contracted with phenylephrine were pre-treated with 20 mM KCl. This resulted in the partial blocking of K^+^ efflux by increasing the extracellular K^+^ concentration to 20 mM and did not change the vasorelaxant effect after cumulative administration of 7-HC (0.001 μM to 300 μM) (effect (300 μM) = 98.4 ± 15.5%, (*n* = 7)) compared with the control (Figure 2B). These results may suggest that the effect of 7-HC could depend, at least in part, on the participation of additional K^+^ channels. Interestingly, the 7-HC-induced vasorelaxation was shifted to the right of the concentration–response curve in the presence of TEA (1 mM), glibenclamide (10 μM), and 4-AP (1 mM) but not BaCl_2_ (30 μM) (Figure 2C–F) compared with the control. These results suggest that the activation of K_ATP_, BK_Ca_, and, to a lesser extent, K_v_, but not K_ir_, plays an important role in the relaxant effect of 7-HC in mesenteric arteries of SHR.

### 2.3. Effect of 7-HC on the CaCl_2_-Induced Concentration–Response Curves

As shown in Figure 3A, the concentration–response curve of CaCl_2_ in a depolarizing medium was shifted to the right in the presence of 7-HC (0.3, 3, 30, and 300 µM) compared with the control. The maximal contraction of CaCl_2_ was significantly (*** *p* < 0.001) attenuated by all concentrations of 7-HC (effect (3 μM) = 31.1 ± 7.3%, (*n* = 5); effect (30 μM) = 36.6 ± 8.3%, (*n* = 5); effect (300 μM) = 43.9 ± 5.8%, (*n* = 5)). These data indicate that the mechanism of action of 7-HC involves reducing the Ca^2+^ influx.

### 2.4. Effect of Store-Operated Calcium Entry (SOCE) Inhibition on the Relaxation Induced by 7-HC in Hypertensive Rats

The vasorelaxation induced by 7-HC (0.001 μM to 300 μM) was reduced in the presence of the SOCE inhibitor SKF-96365 (effect (300 μM) = 52.4 ± 9.3% (*n* = 6)) compared with the control (effect (300 μM) = 100.4 ± 9.8% (*n* = 6)) (Figure 4). This result suggests that the inhibition of Ca^2+^ influx by SOCE is involved in the vasorelaxation mechanism of 7-HC in mesenteric arteries from SHR.

### 2.5. Effect of 7-HC on the Mobilization of Calcium from Intracellular Stores

7-HC (30 μM and 300 μM) markedly reduced the Phe-induced contraction in a Ca^2+^-free Tyrode solution (52.8 ± 14.5% (*n* = 5); 29.1 ± 10.7% (*n* = 5), respectively) compared with the control (100 ± 0.0) (Figure 5A), suggesting that 7-HC may inhibit the release of Ca^2+^ from intracellular stores through the IP3 receptors. Moreover, 7-HC (30 μM and 300 μM) significantly decreased the caffeine-induced contraction in a Ca^2+^-free Tyrode solution (17.3 ± 3.2% (*n* = 5), 11.5 ± 4.3% (*n* = 5), respectively) compared with the control (100 ± 0) (Figure 5D). These data suggest that the mechanism of action of 7-HC involves reducing the release of Ca^2+^ from the intracellular stores, through both the IP3 receptors and ryanodine receptors.

### 2.6. Influence of 7-HC Pre-Incubation on Vascular Reactivity in Hypertensive Rats

The pharmacological efficacy of Phe was significantly increased in superior mesenteric arteries from SHR compared with wistar rats. In contrast, relaxation with the endothelium-independent vasodilator and nitric oxide (NO) donor SNP or acetylcholine was decreased in SHR compared with Wistar rats (Appendix A). These results suggest that the high blood pressure established in SHR promotes vascular dysfunction compared with wistar rats. Phenylephrine’s concentration–response curve was shifted to the right after pre-incubation with 7-HC (100 µM), with a reduction in the pD2 values (*p* < 0.001) compared with the results obtained in the absence of 7-HC (pD_2_ = 6.14 ± 0.05 (*n* = 5, presence of 7-HC); pD_2_ = 6.64 ± 0.07 (*n* = 5, absence of 7-HC)) (Figure 6A,B). However, the maximal response did not change (E_max_ = 176.8 ± 6.1 (*n* = 5); E_max_ = 178.6 ± 4.0 (*n* = 5), respectively, for the presence and absence of 7-HC). In addition, SNP’s concentration–response curve was shifted to the left after pre-incubation with 7-HC (100 µM), with an increase in the pD2 values (*p* < 0.05) compared with the results obtained in the absence of 7-HC (pD_2_ = 8.66 ± 0.09 (*n* = 6, presence of 7-HC); pD_2_ = 9.44 ± 0.31 (*n* = 6, absence of 7-HC)), but no change the maximal response (E_max_ = 129.2 ± 9.0 (*n* = 6); E_max_ = 137.3 ± 19.7 (*n* = 6), respectively, for the presence and absence of 7-HC) (Figure 6C,D). Acetylcholine’s concentration–response curve was shifted to the left after pre-incubation with 7-HC (100 µM), without a significant change in the maximal response and pD2 values compared with the results obtained in the absence of 7-HC (E_max_ = 119.9 ± 4.2 (*n* = 5), pD_2_ = 7.31 ± 0.28 (*n* = 5, presence of 7-HC); E_max_ = 109.9 ± 14.4, pD_2_ = 6.77 ± 0.35 (*n* = 5, absence of 7-HC)) (Figure 6E,F). Taken together, these results suggest that 7-HC can reduce the responsiveness of the mesenteric arteries of SHR to α1-adrenergic agonists and increase the responsiveness to muscarinic agonists and an NO donor, improving vascular endothelium-dependent and independent relaxation.

## 3. Discussion

The present study indicated that the vasorelaxant effect of 7-HC in SHR occurs via an endothelium-independent pathway, which likely consists of the activation of potassium channels (K_ATP_, BK_Ca_, and Kv), and reductions in both calcium influx and intracellular calcium mobilization from the sarcoplasmic reticulum. Additionally, 7-HC reduced the pharmacological potency of α1-adrenergic agonists while increasing the potency of muscarinic agonists and NO donors in mesenteric arteries from SHR, improving vascular endothelium-dependent and independent relaxation. Interestingly, even with the pathophysiological changes generated in the hypertensive condition, 7-HC maintained its antihypertensive vascular effects, supporting its therapeutic potential.

Peripheral vascular resistance plays an essential role in the regulation of blood pressure, and several substances can act on this parameter, contributing to the control of arterial pressure [19]. Vascular endothelial and smooth muscle cells play critical roles in regulating vascular tension [20]. Vascular smooth muscle cells (VSMCs) are exposed to mechanical stress and the pressure of blood flow, and play an important role in maintaining vascular tone and resistance [21]. In addition, the endothelium regulates vascular tone by synthesizing and releasing an array of endothelium-derived relaxing factors, such as nitric oxide and endothelium-dependent hyperpolarization factors, as well as endothelium-derived contracting factors [22,23]. In the current study, when the endothelium of mesenteric artery rings was removed, the vasorelaxant effect of 7-HC was not significantly impaired, indicating that the vasorelaxant activity of 7-HC does not rely on the integrity of the endothelial layer. Thus, subsequent experiments were carried out to investigate 7-HC’s endothelium-independent vasorelaxation mechanisms.

Our research group has previously demonstrated that 7-HC is nontoxic for cardiomyocytes [18,24] and did not have an irreversible action on the arterial contractile machinery in normotensive rats [18]. Our observations corroborate what is shown in the literature, since we also observed that tissue contractility of mesenteric artery rings from SHR was maintained after removing 7-HC from the tissue bath solution, reinforcing the notion that 7-HC has reversible actions on the contractile machinery and probably has no toxic effects on the vascular myocyte in SHR.

Subsequently, we designed experiments to investigate the 7-HC-mediated vasorelaxant mechanisms of action in the arteries of hypertensive animals. It has been reported in the literature that the opening of K^+^ channels promotes hyperpolarization of the membrane potential and closure of voltage-gated channels, which decreases the entry of Ca^2+^ and causes vasorelaxation [25,26,27]. In our study, the increase in extracellular K^+^ concentrations (from 4 to 20 mM) did not significantly attenuate 7-HC induced vasorelaxation, which diverges from the results obtained by Alves and colleagues (2020) [18] in the mesenteric artery from normotensive rats. Such differences in the tissue response can be attributed to the altered expression or activity of K^+^ channels, as in the experimental model of hypertension [28]. For example, an attenuation of the K_v_ channel’s activity during hypertension has been described in VSMCs from rat mesenteric arteries [29,30,31]. However, 7-HC vasorelaxation was attenuated considerably when the rings were exposed to Tyrode’s solution containing 80 mM KCl, suggesting the participation of K^+^ channels in 7-HC-induced vasorelaxation. Furthermore, the higher concentration of K^+^ in the solution (>20 mM) could influence the response to 7-HC in the superior mesenteric artery rings of SHR. Thus, the involvement and activation of distinct types of K^+^ channels in 7-HC-induced vasorelaxation were investigated.

K^+^ channels play an important role in the regulation of vascular tone, since the activation of K^+^ channels is known to promote blood vessel relaxation through membrane hyperpolarization, leading to a decreased probability of voltage sensitive opening of the calcium channels, thereby lowering intracellular calcium levels [32,33]. The regional vasodilator profile of K^+^ channels’ activation is heterogeneous and, in patients, only pinacidil, minoxidil, and diazoxide are currently used as antihypertensive drugs.

Minoxidil is an arteriolar vasodilator, and the antihypertensive activity of minoxidil is due to its sulfate metabolite. Minoxidil sulfate opens the adenosine tri-phosphate-sensitive potassium channels in vascular smooth muscle cells [34]. Pinacidil, another known potassium channel opener, also produces vasodilation comparable with that of minoxidil [35].

Disorders in the structure and function of K^+^ channels contribute to the pathophysiological conditions of some vascular diseases, including hypertension [31]. K^+^ currents are essential for preserving membrane stability by suppressing the membranes’ excitability. However, in the hypertensive condition, the K^+^ currents are lower than in the normotensive state [36]. Thus, a reduction in the potassium efflux current can affect the excitability of vascular cells, causing membrane depolarization and resulting in hypertension [37].

There are different subtypes of K^+^ channels. Calcium-activated K^+^ (K_Ca_) channels are the most abundant K^+^ channels found in the VSMC and are activated by increases in the intracellular Ca^2+^ concentration [25]. In particular, BK_Ca_ channels in the VSMC are especially important for regulating vascular tone and are responsible for a significant proportion of potassium efflux [38]. In our study, to evaluate the participation of BK_Ca_ channels in the 7-HC-mediated relaxation of SHR mesenteric rings, assays were performed utilizing 1 mM TEA, as TEA blocks BK_Ca_ channels at this concentration [39], and the vasorelaxant effect of 7-HC was significantly reduced, suggesting the participation of these channels in the mechanism of 7-HC.

A similar response was observed in experiments with mesenteric rings pre-incubated with glibenclamide (10 µM) and 4-AP (1 mM), which are blockers of the ATP-sensitive K^+^ channels and the voltage-sensitive K^+^ channels, respectively. In both cases, the concentration–response curve was shifted to the right, indicating a decrease in the pharmacological potency of 7-HC when challenged with the respective blockers, suggesting the participation of ATP-sensitive K^+^ channels, or voltage-sensitive K^+^ channels.

The same response pattern was not observed when the mesenteric arteries were preincubated with BaCl_2_ (30 µM), an inward-rectifier blocker of K^+^ channels. In this case, the vasorelaxant effect of 7-HC remained unaltered in the presence or absence of the blocker, suggesting that these channels may not participate in 7-HC’s mechanism of action in the mesenteric arteries of SHR. By contrast, Alves and colleagues (2020) [18] proposed that 7-HC caused the opening of BaCl_2_-sensitive inward rectifying K^+^ channels in the mesenteric rings from normotensive rats. Although there is no difference in the expression of inward rectifying K^+^ channels in SHR and normotensive Wistar-Kyoto rats, the inward rectifying K^+^ channels in SHR are less involved in the spread of Ach-induced hyperpolarization compared with normotensive rats [40]. Therefore, the different responses observed following the application of 7-HC in both strains may rely on the different physiological K_ir_ responses in normotensive rats and SHR. In addition to the elevated vascular tone observed in humans and in experimental hypertension models, such as SHR [41], there have also been reports that these cases involve alterations of various families of Ca^2+^ and/or K^+^ channels in the expression or functionality in the VSMC, consisting of a variety of subtypes including voltage-gated Ca^2+^ channels (Ca_V_) [42], voltage-dependent K^+^ channels (K_v_) [43], and the broad-conductance Ca^2+^-activated K^+^ channels (BK_Ca_) [44,45]. This demonstrates the relevance of this study, as we observed the effect of 7-HC on the BK_Ca_ and K_v_ channels, which are known to participate in the pathophysiology of hypertension.

Calcium channel blockers (CCBs) were shown to lower elevated blood pressure more than 50 years ago. The drug now known as verapamil was described in 1962 as increasing renal blood flow and lowering blood pressure in hypertensive but not in normotensive subjects [46], and is still used in the treatment of hypertension. CCBs are recommended in the Eighth Joint National Committee (JNC8) guidelines to be used as a first-line treatment alone or in combination with other antihypertensives in all patients with hypertension regardless of age and race, except for patients with chronic kidney disease where angiotensin-converting enzyme inhibitors and angiotensin receptor blockers inhibitors are the recommended first-line treatment [47].

The scientific literature has reported significant abnormalities in calcium signaling in the VSMCs of SHR compared with normotensive rats [48,49,50,51,52]. Both extracellular Ca^2+^ influx through Ca_V_ or store-operated calcium entry (SOCE) can increase intracellular calcium. Previous studies have shown that the current densities of Ca_V_ are elevated in mesenteric VSMCs of SHR compared with WKY rats [53,54,55]. This elevation in the Ca^2+^ currents may be due, at least in part, to upregulation of the expression of the pore-forming α_1 C_ subunit of the Ca_V_ channel in the mesenteric artery from SHR [50]. SOCE increases the intracellular Ca^2+^ concentration, which is essential for producing the intracellular calcium signals needed to regulate vascular tone [56,57]. 7-HC (0.3, 3, 30, and 300 µM) inhibits Ca^2+^ influx-mediated vasoconstriction in mesenteric rings pre-incubated with a depolarizing solution, suggesting that 7-HC may inhibit Ca_V_.

In addition, experiments were performed to assess the involvement of SOCE in the relaxation induced by 7-HC, using the nonspecific channel blocker SKF 96365. The results indicated that the 7-HC-mediated relaxation was reduced in the presence of a SOCE inhibitor, suggesting that 7-HC-induced vasorelaxation may be due to a reduction in the influx of Ca^2+^ through the SOC.

Inositol 1,4,5-trisphosphate receptors (IP3 R) and ryanodine receptors are the channels that most commonly mediate the release of Ca^2+^ from intracellular stores [58,59]. When activated by caffeine, the ryanodine receptor works through a Ca^2+^-induced Ca^2+^ release mechanism (CICR). Alternatively, Phe, an α1-adrenergic agonist, enhances extracellular Ca^2+^ entry by activating both ROCCs and Ca_V_ [60], as well as inducing the release of Ca^2+^ from IP3 R. However, the vasoconstriction induced by Phe in the Ca^2+^-free medium is due to endo/sarcoplasmic reticulum (ER/SR) Ca^2+^ release through the IP3-sensitive Ca^2+^ channels. In our experiments, this vasoconstriction was inhibited by 7-HC (30, 300 µM) in a concentration-dependent manner. This suggests that the endothelium-independent relaxation evoked by 7-HC may inhibit the mobilization of Ca^2+^ from intracellular stores through the IP3-sensitive Ca^2+^ channels. Furthermore, 7-HC (30, 300 µM) also decreases caffeine-induced contraction in endothelium-denuded arteries, suggesting the potential involvement of ryanodine receptors induced by 7-HC. These results indicate that the vasorelaxant response of 7-HC may also involve, at least in part, the inhibition of extracellular Ca^2+^ entry by Ca_V_ and the inhibition of Ca^2+^ released from the internal store by the IP3 receptors and ryanodine receptors, leading to a decrease in intracellular Ca^2+^.

Hypertension is directly related to endothelial dysfunction and low NO bioavailability due to the decreased NO production and/or increased NO degradation caused by oxidative stress [61]. Therefore, this may lead to increased contractile and/or decreased vasodilatory responses. In the present study, pre-incubation with 7-HC affected contractile responses in the mesenteric arteries of hypertensive rats. We observed that the pharmacological potency of the α1-adrenergic agonist was reduced in the presence of 7-HC, reinforcing that in a hypertensive state, where adrenergic responsiveness is increasing, 7-HC may be a therapeutic alternative.

The ACh-induced relaxation in isolated arteries relies on muscarinic receptors in the endothelial cells to improve several endothelial factors [62]. At the same time, SNP, a NO donor, directly induces vasorelaxation in the vascular smooth muscle, mainly by activating soluble guanylate cyclase [63]. It has been established that the reduced relaxation response to ACh or SNP in SHR, due to endothelial dysfunction, is caused by oxidative stress [64,65], decreased production and bioavailability of NO [66,67,68], eNOS uncoupling [64,69], and hyper-responsiveness to contracting agents such as angiotensin II [70,71,72] and endothelin-1 [73]. Additionally, ACh or SNP-mediated vasorelaxation was diminished in SHR compared with normotensive rats [67,74]. Our study showed that 7-HC improved the endothelium-dependent and independent vasorelaxation evoked by ACh and SNP, respectively, suggesting that 7-HC increased the arterial sensitivity to vasorelaxant signaling pathways, which makes this coumarin a potential drug for treating diseases with endothelial dysfunction. However, further studies are needed to assess whether 7-HC could prevent chronical endothelial dysfunction, improve NO bioavailability, and prevent vascular remodeling in hypertensive conditions.

## 4. Material and Methods

### 4.1. Animals

Male spontaneously hypertensive rats (SHR) at the age of 12–15 weeks, weighing 200 to 300 g, were used in all experiments. The animals were supplied by the animal facility of the Neuroscience Laboratory at the Institute of Health Sciences at the Federal University of Bahia and kept under a controlled temperature (23 ± 1.0 °C) and a light–dark cycle of 12 h (6:00 a.m. to 6:00 p.m.), with free access to food and water. The tests were performed in accordance with the guidelines for the care and use of laboratory rats adopted by the National Council for Animal Experiments Control (CONCEA–Brazil). It was approved by the Ethics Committee on Animal Use from the Institute of Health Sciences, Federal University of Bahia (CEUA/UFBA no. 130/2017).

### 4.2. Drugs and Solutions

The drugs used in this study were: L-phenylephrine hydrochloride (Phe), acetylcholine chloride (ACh), sodium nitroprusside (SNP), barium chloride (BaCl_2_), 4-aminopyridine (4-AP), tetraethylammonium (TEA), glibenclamide (Glib), caffeine, and cremophor, all acquired from Sigma-Aldrich (Sigma Chemical Co., Saint Louis, MO, USA). All drugs were dissolved in distilled water. K^+^-depolarizing solutions (20 and 80 mM KCl) were prepared by replacing equimolar NaCl with 20 or 80 mM of KCl. The 7-hydroxycoumarin (7-HC) was solubilized in cremophor and diluted to the desired concentrations with distilled water, as described by Alves and colleagues (2020) [18]. Tyrode’s physiological solution was used in all arterial experiments, with the following composition at 37 °C (in mM): NaCl, 158.3; KCl, 4.0; CaCl_2_, 2.0; MgCl_2_, 1.05; NaH_2_PO_4_, 0.42; NaHCO_3_, 10.0; glucose, 5.6. The reagents were acquired from Sigma (Sigma-Aldrich, St. Louis, MO, USA) or Vetec (Vetec, Rio de Janeiro, Brazil).

### 4.3. Isolation of the Superior Mesenteric Arteries

SHR were euthanized in a CO_2_ chamber, and the middle portion of the superior mesenteric arteries was removed, as previously described by Silva and colleagues (2011) [75]. The mesenteric rings (2.0 mm) were stabilized with an optimal resting tension of 7.5 mN for 60 min in an organ bath with physiological Tyrode’s solution (at 37 °C and pH 7.4) and gassed with a carbogenic mixture (95% O_2_ and 5% CO_2_). The isometric tension was recorded by a force transducer (Insight, Ribeirão Preto, Sao Paulo, Brazil) coupled with an amplifier-recorder (Insight, Ribeirão Preto, Sao Paulo, Brazil) and a personal computer equipped with data acquisition software. The presence of a functional endothelium was assessed by the ability of ACh (1 μM) to induce at least a 90% relaxation response in mesenteric artery rings pre-contracted with Phe (1 μM). Rings that relaxed by less than 10% were considered endothelium-denuded/damaged.

### 4.4. Effect of 7-HC on the Vasculature

After the stabilization period, two successive contractions of similar magnitude were induced with Phe (1 μM), an α1-adrenergic agonist, in endothelium-intact and endothelium-denuded rings. In the tonic phase of the second contraction, increasing concentrations of 7-HC (0.001–300 μM) were cumulatively added to the organ bath, and the effects were compared. Another set of experiments investigated the reversibility of the effects of 7-HC. In these assays, a contraction was performed with Tyrode’s solution, with 80 mM KCl, in absence of 7-HC; after the cumulative addition of 7-HC, the tissue was washed, stabilized, and then contracted again with Tyrode’s solution with 80 mM KCl.

### 4.5. Evaluation of K^+^ Channel Activity in the 7-HC-Induced Vasorelaxation Response

Endothelium-denuded ring preparations were pre-incubated with Tyrode’s solution containing 80 mM KCl, or Phe (1 μM) in Tyrode’s solution with 20 mM KCl, and the relaxing effect induced by 7-HC (0.001–300 μM) was obtained. In addition, the involvement of K^+^ channels in the vasorelaxant response induced by 7-HC was evaluated following pre-incubation of the endothelium-denuded rings for 30 min with the following different pharmacological agents: TEA (1 mM), a nonselective blocker of large-conductance Ca^2+^-activated K^+^ (BK_Ca_) channels [76]; 4-AP (1 mM), a voltage-operated K^+^ channel (K_v_) blocker [77]; BaCl_2_ (30 μM), a blocker of inward rectifying K^+^ channels (K_ir_) [78]; and glibenclamide (10 μM), a specific ATP-sensitive K^+^ (K_ATP_) channel subtype-selective blocker [79].

### 4.6. Investigation of the Effects Induced by 7-HC in Ca^2+^ Influx

The method was adapted from that described by Alves et al. 2020 [18]. To investigate the influence of 7-HC on the influx of Ca^2+^ in endothelium-denuded rings, cumulative concentrations of CaCl_2_ (100–10,000 μM) were added to a depolarization medium of Ca^2+^-free Tyrode’s solution containing 80 mM KCl (for 15 min) in the absence (control) or presence of different concentrations of 7-HC (3, 30 and 300 µM). Another experiment was performed to investigate the influx of Ca^2+^ through SOCE (store-operated calcium entry), where experiments were performed using SKF 96365 (10 µM, a SOCE blocker). Endothelium-denuded rings were pre-incubated with SKF for 30 min; subsequently, Phe (1 μM) was added.

### 4.7. Investigation of 7-HC-Induced Mobilization of Calcium from Intracellular Stores

After the stabilization period, vessels were pre-contracted with Tyrode’s solution containing 80 mM KCl for 3 min, then washed with a Ca^2+^-free Tyrode solution containing EGTA 1 mM, followed by the addition of 1 μM Phe (in an organ bath at 37 °C) or 20 mM caffeine (in an organ bath at 23 °C). Rings were rinsed with Tyrode’s physiological solution, and Tyrode’s solution containing 80 mM KCl was added for 3 min to load the Ca^2+^ stores within the vascular smooth muscle. This procedure was repeated until two transient contractions of similar magnitude had been obtained. Next, this experimental protocol was repeated by incubating 7-HC (10 and 100 μM) in Ca^2+^-free Tyrode’s solution for 5 min, then applying Phe or caffeine, as previously described [80]. The role of 7-HC on the mobilization of calcium from the intracellular stores was assessed by comparing Phe or caffeine-mediated contraction in the absence or presence of 7-HC via the inositol 1,4,5-triphosphate (IP3) and ryanodine receptors, respectively.

### 4.8. Influence of 7-HC on the Vascular Reactivity in Hypertensive Rats

To investigate the influence of 7-HC on arterial reactivity in SHR, cumulative concentration–response curves were obtained for α1- adrenergic (Phe), muscarinic (ACh), and nitric oxide (NO) donor (SNP) agonists. After the stabilization period, the arterial rings were contracted with Phe (1 μM) and then assessed to determine the relaxant effect induced by ACh (10^−11^ to 3 × 10^−6^ M) or SNP (10^−11^ to 3 × 10^−6^ M). The contractile response to Phe (10^−10^ to 3 × 10^−5^ M) was also assessed. Next, the tissues were washed and incubated for 30 min with 7-HC (100 µM) and then challenged again with the same cumulative curves in the presence of 7-HC.

### 4.9. Data Analysis

Data are presented as the mean ± standard error of the mean (S.E.M.), and *n* represents the number of mesenteric artery rings obtained from different animals. All calculations were performed using Prism software version 5.0 (GraphPad Software Inc., CA, USA). The pharmacological potency, expressed as pD_2_ (pD_2_ = −logEC_50_), was defined in the experiments where E_max_ (the maximum effect generated by the agonist) could be determined. Data were analyzed with unpaired *t*-tests, two-tailed tests, and 95% confidence intervals. One-way ANOVA followed by Bonferroni’s post-test was used when data were compared among more than two groups. Statistical significance was set at *p* < 0.05.

## 5. Conclusions

The results demonstrated that 7-HC induces vasorelaxation in the mesenteric artery from animals with essential hypertension involving the activation of K^+^ channels, inhibition of the Ca^2+^ influx through Cav and SOC, and a reduction in the release of calcium from ryanodine- and IP3-sensitive intracellular stores. Furthermore, these results demonstrated that the vasorelaxant effect of 7-HC on the vascular smooth muscle cells from the superior mesenteric artery of SHR is independent of endothelium-derived relaxing factors, most likely through the activation of the K_ATP,_ BK_Ca,_ and K_v_ channels. Additionally, 7-HC can attenuate calcium influx, which leads to a reduction in the mobilization of intracellular calcium. Furthermore, 7-HC also seems to be able to improve endothelial function and the relaxing ability of vascular smooth muscle, attenuating the vascular dysfunctions associated with hypertension. Thus, 7-HC is a promising compound with direct vascular activity and may be considered a potential strategy for the management of hypertension.

## Figures and Tables

**Figure 1 molecules-27-07324-f001:**
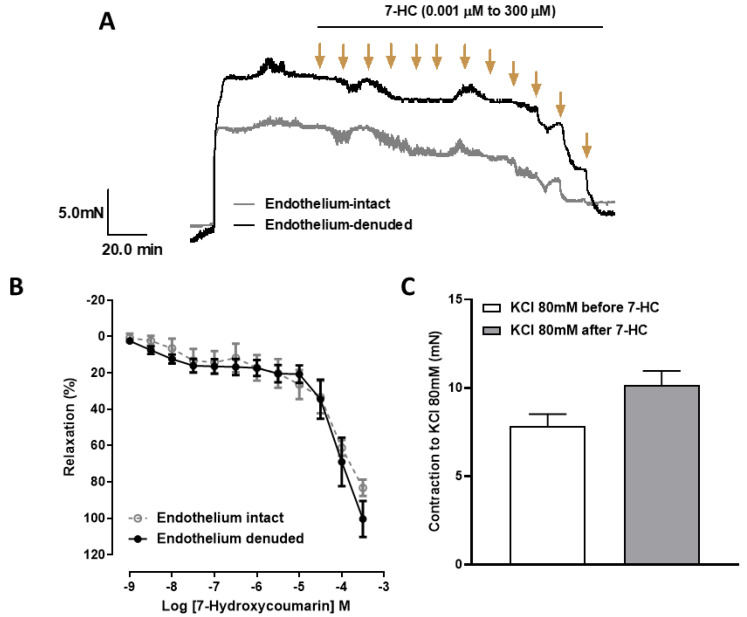
Vasorelaxation effects of 7-HC in the superior mesenteric artery from SHR. (**A**) Representative original recordings of the effects of 7-HC in endothelium-intact and endothelium-denuded mesenteric arteries of rats. (**B**) Relaxation responses induced by 7-HC (0.001–300 μM) in endothelium-intact (○, *n* = 6) and endothelium-denuded (●, *n* = 6) rat mesenteric arterial rings, pre-contracted with Phe (1 μM). (**C**) Bar graphs showing the contraction induced by Tyrode’s solution containing 80 mM KCl before and after the 7-HC concentration–response curve. Results are expressed as means ± S.E.M. Statistical analysis was performed using unpaired Student’s *t*-tests.

**Figure 2 molecules-27-07324-f002:**
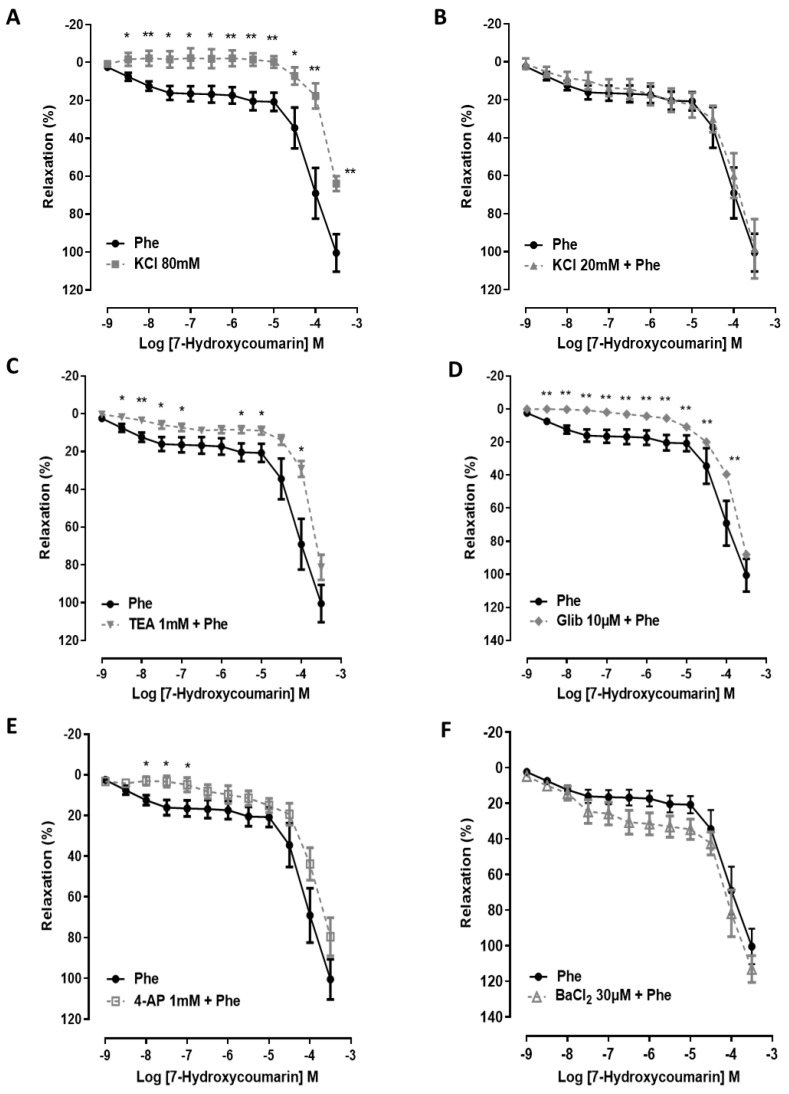
K^+^ channels’ involvement in 7-HC-induced relaxation in the superior mesenteric artery of SHR. Vasorelaxation responses induced by 7-HC (0.001 μM–300 μM) in denuded mesenteric artery rings of SHR pre-contracted with Phe (1 μM) (●, *n* = 6), 80 mM KCl (■, *n* = 5) (**A**), 20 mM KCl + Phe (▲, *n* = 5) (**B**), 1 mM TEA (▼, *n* = 5) (**C**), 10 µM glibenclamide (♦, *n* = 5) (**D**), 1 mM 4-AP (□, *n* = 5) (**E**), and 30 µM BaCl_2_ (∆, *n* = 5) (**F**). Results are expressed as means ± S.E.M. Statistical analysis was performed using unpaired Student’s *t*-tests. * *p* < 0.05 or ** *p* < 0.01 versus Phe.

**Figure 3 molecules-27-07324-f003:**
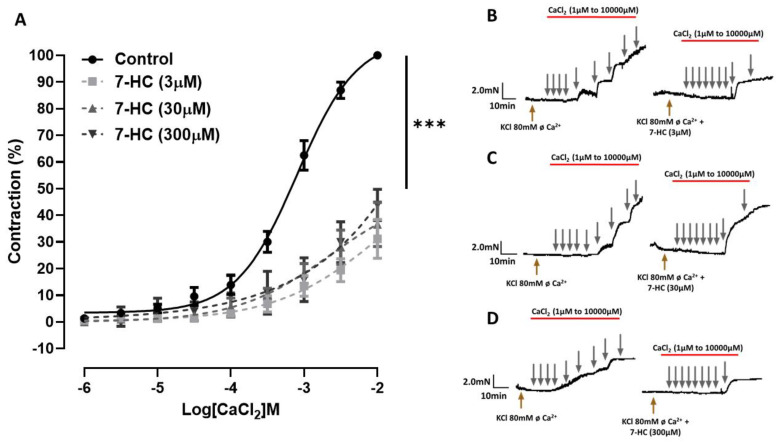
Effect of 7-HC on Ca^2+^ influx. (**A**) Concentration–response curve of CaCl_2_ on mesenteric artery segments from SHR without endothelium, in the absence (●, *n* = 15) or in the presence of 7-HC (■, 3 μM, *n* = 5; ▲, 30 μM, *n* = 5; ▼, 300 μM, *n* = 5). Representative original recordings of the effects of 3 μM 7-HC (**B**), 30 μM 7-HC (**C**), and 300 μM 7-HC (**D**) on isolated mesenteric artery rings pre-incubated with 80 mM calcium-free KCl. The results are expressed as means ± S.E.M. Statistical analysis was performed using one-way ANOVA followed by Bonferroni’s post-test. *** *p* < 0.001 versus the control.

**Figure 4 molecules-27-07324-f004:**
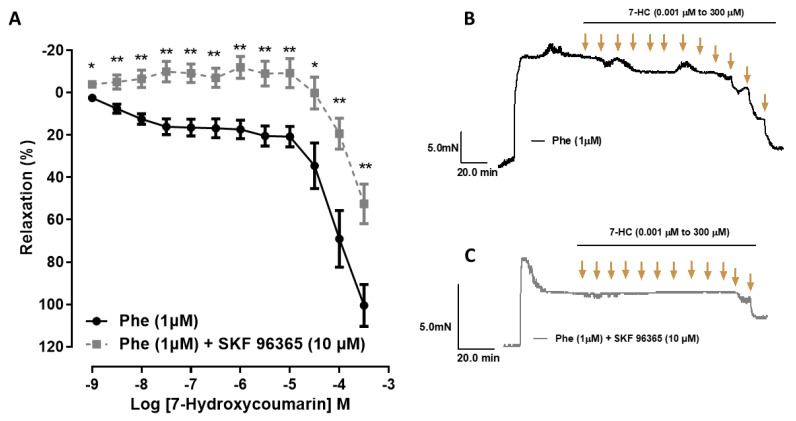
The vasorelaxant effect of 7-HC in the presence of SOCE inhibition in mesenteric artery of SHR. (**A**) Relaxation responses induced by 7-HC (0.001–300 μM) on the isolated endothelium-denuded mesenteric artery rings, pre-contracted with Phe, in the presence or absence of SKF-96365 (10 μM) (■, *n* = 5) compared with the control (●, *n* = 6). Representative original recordings of the effects of 7-HC in isolated mesenteric artery rings, pre-contracted with phenylephrine (**B**) or SKF-96365 (10 µM) (**C**). Results are expressed as means ± S.E.M. Statistical analysis was performed using unpaired Student’s *t*-tests. * *p* < 0.05 or ** *p* < 0.01 versus Phe.

**Figure 5 molecules-27-07324-f005:**
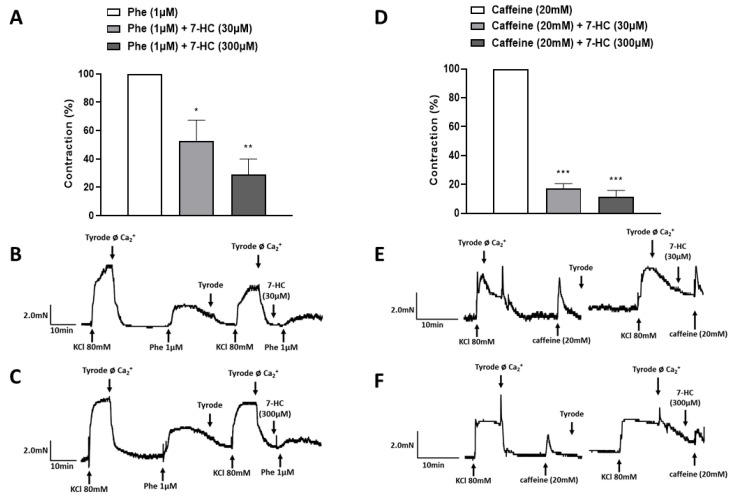
Effect of 7-HC on the mobilization of intracellular Ca^2+^. Bar graphs showing the effect of 7-HC (30 μM and 300 μM) on the Ca^2+^ mobilization from phenylephrine-sensitive (**A**) and caffeine-sensitive (**D**) intracellular stocks in a Ca^2+^-free Tyrode solution. Representative recordings of the effects of 30 μM and 300 μM 7-HC on isolated mesenteric artery rings incubated with 1 μM phenylephrine (**B**,**C**) or 20 mM caffeine (**E**,**F**) in a calcium-free solution. The results are expressed as means ± S.E.M. Statistical analysis was performed using one-way ANOVA followed by Bonferroni’s post-test. * *p* < 0.05, ** *p* < 0.01, or *** *p* < 0.001 versus Phe (**A**) or caffeine (**D**).

**Figure 6 molecules-27-07324-f006:**
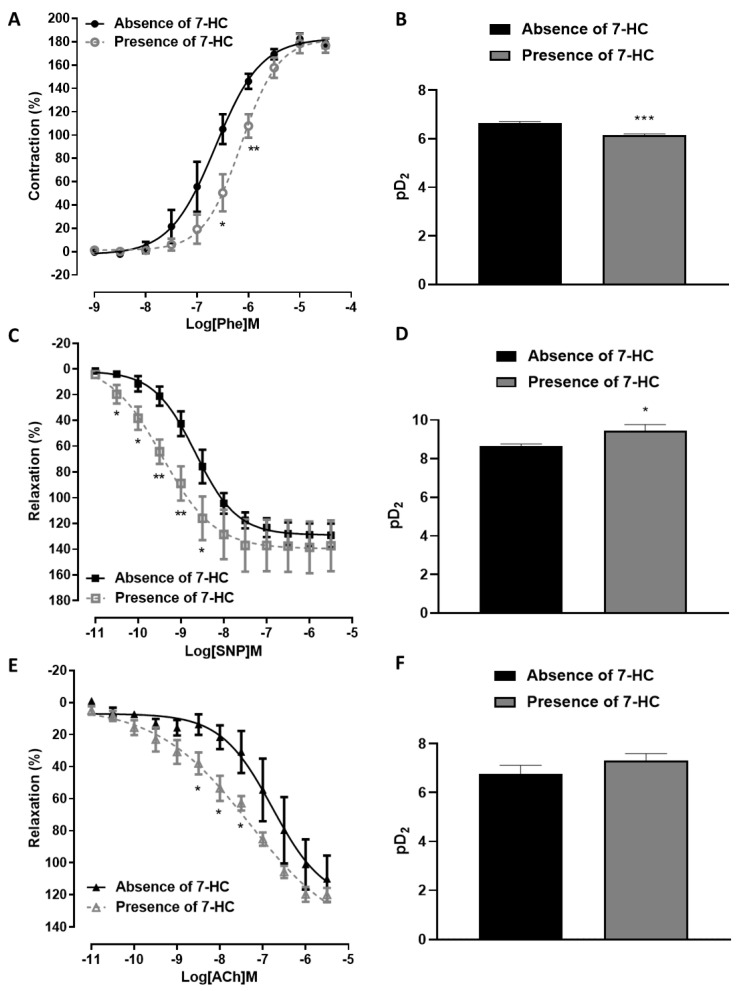
7-HC modulates the vascular reactivity of the mesenteric artery in SHR. (**A**) Contraction responses induced by Phe (10^−10^ to 3 × 10^−5^ M, cumulatively) in the absence of 7-HC (●, *n* = 5) and the presence of 7-HC (○, *n* = 5). (**B**) Bar graph with the values of pD_2_ from Phe. (**C**) Relaxation responses induced by SNP (10^−11^ to 3 × 10^−6^ M, cumulatively) in the absence of 7-HC (■, *n* = 5) and the presence of 7-HC (□, *n* = 5). (**D**) Bar graph with the values of pD_2_ from SNP. (**E**) Relaxation responses induced by Ach (10^−11^ to 3 × 10^−6^ M, cumulatively) in the absence of 7-HC (▲, *n* = 5) and the presence of 7-HC (∆, *n* = 5). (**F**) Bar graph with the values of pD_2_ from ACh. The results are expressed as means ± S.E.M. Statistical analysis was performed using unpaired Student’s *t*-tests. * *p* < 0.05, ** *p* < 0.01 or *** *p* < 0.001 versus the absence of 7-HC.

## Data Availability

Not applicable.

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
