# Peer review of "7-Hydroxycoumarin Induces Vasorelaxation in Animals with Essential Hypertension: Focus on Potassium Channels and Intracellular Ca2+ Mobilization"

_molecules, 2022, doi:10.3390/molecules27217324_

Round 1

Reviewer 1 Report

The paper from Jesus at al it’s a continuation of a study previously done by the same laboratory where they inspected the effect of 7-HC in normotensive rats. Now they inspect the effect of 7-HC in spontaneously hypertensive rats. The study seems interesting and relevant, however there are some concerns of mine that hopefully the authors can address.

Title

The authors cannot conclude that 7-HC improves the vascular dysfunction in animals with EH, as this compound was not given in vivo, but ex vivo. So, this needs to be rephrased to better represent the results of the study.

Introduction

Please introduce more the relevant antihypertensive treatment, as what is their current mode of action and how that correlates or not with your treatment of interest. Also, this should be addressed in the discussion.

Methods and Results

Was the hypertension confirmed in these rats?  If not, why?

Why only male rats were used? CVD and hypertension are prominent in both sexes.

Did the authors used a control of non-hypertensive rats?

Author Response

Reply to referee 1 is attached.

Reviewer 2 Report

Rafael Leonne C. Jesus et al. studied the vasorelaxant effect of 7-HC on the superior mesenteric artery of spontaneously hypertensive rats. Likewise, they propose a mechanism by which this molecule generates such effect in this experimental model. In general, several aspects of the manuscript must be clarified, and althougth the subject is interesting, it has several errors thus authors should carefully revised it and corrected it. Also the narrative of the text need to be improved to make the manuscript more appealing to the journal readers. Some points are listed below:

MAJOR POINTS

1.   The title of the manuscript needs to be described more precisely because it does not clearly reflect what was done in the study. The authors evaluated the vasorelaxant effect of 7-HC in an ex vivo experimental model of the superior mesenteric artery of spontaneously hypertensive rats (SHR).

2.     How was improvement in vascular dysfunction assessed? What was your control?

3.   It is important to substantiate why the superior mesenteric artery was used, compared to another segment of the vasculature such as the aorta. Some studies have reported that the contraction and relaxation patterns may differ between both structures (doi: 10.1038/s41598-019-43193-8).

4.  The main objective of the study was to evaluate whether 7-HC generated vasorelaxation in an ex vivo model of hypertension. So, was it initially established that compared to normotensive rats, the mesenteric arteries of the HSR in the study responded differently to the substances studied (Phe, KCl, ACh, etc.)? This fact is important to understand that the experimental model was correctly established (this information should be mentioned in supplementary material).

5.      The age of the HSR must be described in the manuscript

6.     Regarding the statistical analysis section, the following should be described and clarified:

-Were the dose-response curves obtained by non-linear regression?

-The statistical analysis model for the dose-response curves was unpaired Student's t-test. How were the multiple comparisons made? One way to analyze these curves is through a two-way ANOVA (Concentration vs. % effect) followed by a multiple comparison test such as Bonferroni (XY table, in GraphPad Prism).

-The authors calculated the pD2 value, but they do not explain what this value means or how they calculated it. Are the authors referring to what we currently know as pEC50?, a parameter that explains the sensitivity (density of the receptor in a tissue) against an agonist.

-How were figures 6B, 6D and 6F constructed? Did the authors obtain various pD2 values ​​with the statistical program and represent the average? or did they represent the single pD2 value using bars?

The study has important findings for the pharmacological field, but an important limitation is that the results obtained are only compared in the context of the hypertensive rat. It would have been interesting to know if 7-HC relaxes the mesenteric artery of the SHR to a lesser, similar or greater degree extent than in the mesenteric artery of normotensive rats.

MINOR POINTS

-     Line 79, change (Figure 1A) to (Figure 1A and 1B).

-   Line 81, Figure 1B does not correspond to what is stated in this sentence.

-    In the material and methods section (line 359) it is indicated that the adjustment tension was 0.75 g, and in the results section the original records correspond to units of mN. Homogenize or describe the equivalences of force.

- Write properly the abbreviated names of the ion channels (BKca, Kir, Kv, KATP) throughout the entire manuscript.

- Reference 4 should be updated to the most current report (doi: 10.1097/HJH.0000000000001940 or 10.1093/eurheartj/ehy339).

-  In Figure 1A, 4B and 4C, indicate by means of arrows the addition of each concentration of 7-HC in the original record of the experiment.

-     In Figure 1C, change 7HC to 7-HC.

-    In Figure 2F, change BaCl2 to BaCl2.

-    In Figure 3A, change CaCl to CaCl2.

- In the description of Figure 3, calcium-free KCl 60 mM is described, whereas in the same figure it is shown as calcium-free KCl 80 mM. The Alves et al. indicates that they used a 60 mM calcium-free KCl solution and in this manuscript it was 80 mM calcium-free KCl (line 392). If it was an adaptation or modification of said reference, it should be reported.

- Figure 4B and Figure 4C, if possible, the records should be presented on the same force scale for an adequate comparison.

- The dose-response curves in figures 1B, 2(A-F) and 4A of this study do not seem to represent a logarithmic trend as claimed. Verify.

-  Line 147, change Figure 3B to Figure 5A.

-   Line 150, change Figure 3C to Figure 5D.

-   Delete the name of the figures from the discussion.

- Discussion: L-type calcium channels (Cav1.2) are voltage dependent calcium channels (CaV), so the term should be standardized.

Author Response

Reply to referee 2 is attached.

Round 2

Reviewer 1 Report

No further comments to the authors

Reviewer 2 Report

Rafael Leonne C. Jesus et al. adequately and accurately answered the comments and suggestions made by this reviewer. The present study and the previously carried out (10.1016/j.ejphar.2020.173525) complement each other and jointly explain how 7-HC relaxes the mesenteric artery of Wistar and SHR rats. Future studies could include both models in the same article (such as Figure S1), to understand if 7-HC improves the response in SHR to a similar degree in wistar (normotensive) rats.

In my opinion, the manuscript is ready for publication after correcting the following minor points:

- "superior mesenteric artery" should be included in the keywords.

- Material and methods: There is a space between lines 379 and 380

- Line 300: change "voltage-gated Ca2+ channels" to voltage-gated Ca2+ channels

- Line 382: change BaCl2 to BaCl2

- Line 480: change 4 AP to 4-AP